# Urethral Injury in Rectal Cancer Surgery: A Comprehensive Study Using Cadaveric Dissection, Imaging Analyses, and Clinical Series

**DOI:** 10.3390/cancers15204955

**Published:** 2023-10-12

**Authors:** Pere Planellas, Lídia Cornejo, Aram Ehsan, Francisco Reina, Nuria Ortega-Torrecilla, Eloy Maldonado, Antoni Codina-Cazador, Margarita Osorio, Ramon Farrés, Anna Carrera

**Affiliations:** 1Department of General and Digestive Surgery, University Hospital of Girona, 17007 Girona, Spain; nortega.girona.ics@gencat.cat (N.O.-T.); emaldonadom.girona.ics@gencat.cat (E.M.); rfarres.girona.ics@gencat.cat (R.F.); 2Department of Medical Sciences, Faculty of Medicine, University of Girona, 17190 Girona, Spain; 3Girona Biomedical Research Institute (IDIBGI), 17190 Girona, Spain; lcornejo@idibgi.org (L.C.); codinac@telefonica.net (A.C.-C.); 4Department of Diagnostic Imaging Institute (IDI), University Hospital of Girona, Girona Biomedical Research Institute (IDIBGI), 17007 Girona, Spain; eehsan.girona.ics@gencat.cat (A.E.);; 5Clinical Anatomy, Embryology and Neurosciences Research Group (NEOMA), Medical Sciences Department, University of Girona, 17007 Girona, Spain; francisco.reina@udg.edu (F.R.); anna.carrera@udg.edu (A.C.)

**Keywords:** urethral injury, rectal cancer, colorectal surgery, pelvic anatomy

## Abstract

**Simple Summary:**

Urethral injuries in men undergoing abdominoperineal resection or TaTME for rectal cancer are uncommon but devastating. The review of images obtained from cadavers and the examination of MRI images of male pelvises allows us to delve deeper into anatomical knowledge in order to achieve a better understanding and prevent urethral injury during rectal cancer surgery. Measurements have described the critical point for injury lies 2–7.3 cm from the anal margin, with a 0.2–2.3 cm distance between rectum and membranous urethra.

**Abstract:**

Male urethral injury during rectal cancer surgery is rare but significant. Scant information is available about the distances between the rectourethral space and neighboring structures. The aim of this study is to describe the anatomical relations of the male urethra. This three-pronged study included cadaveric dissection, retrospective MRI analysis, and clinical cases. Measurements included the R-Mu distance (shortest distance between the rectum and the membranous urethra), R-Am distance (distance from the anterior rectal wall to anal margin nearest to the membranous urethra), and the anal canal–rectum axis angle. The clinical study analyzed the incidence of urethral injury and associated factors among 244 consecutive men from January 2016 to January 2023. The overall incidence of urethral injury in our series was low (0.73%), but in men with tumors < 10 cm from the anal margin, it was 4% in abdominoperineal resection and 3.2% in TaTME. On preoperative MRI, the median R-Mu distance was 1 cm (IQR, range, 0.2–2.3), the median R-Am distance was 4.3 cm (range, 2–7.3), and the median anorectal angle was 128° (range, 87–160). In the cadaveric study (nine adult male pelvises), the mean R-Mu distance was 1.18 cm (range 0.8–2), and the mean R-Am distance was 2.64 cm (range 2.1–3). Avoiding urethral injury is crucial. The critical point for injury lies 2–7.3 cm from the anal margin, with a 0.2–2.3 cm distance between the rectum and the membranous urethra. Collaborating with anatomists and radiologists improves surgeons’ anatomy knowledge.

## 1. Introduction

Despite advances in rectal cancer surgery, the complexity of the procedures involves a high risk of short-term complications that can lead to significant long-term sequelae affecting other organs, pelvic nerves, blood vessels, and, in men, urethral damage [1].

Urethral complications in rectal cancer surgery predominantly result from damage to the membranous urethra during abdominoperineal resection (Miles procedure) or transanal total mesorectal excision (TaTME) [2]. Data on urethral injury during abdominoperineal resection in males are scant, but classical studies estimate its incidence at about 2% [3,4]. The implementation of TaTME has been associated with an increase in urethral injury during rectal cancer surgery, with a reported incidence of 0.8% according to the International TaTME Registry [5].

Patient-related factors associated with urethral injuries during colorectal surgery include prior pelvic surgery, radiation, inflammatory bowel disease, infectious processes, and urogenital abnormalities. A thorough understanding of urogenital anatomy and the ability to identify and repair potential injuries are essential for preventing and managing these injuries [6,7].

The membranous urethra, situated between the prostate gland and the bulb of the penis, is the shortest and narrowest segment of the male urethra. It is enveloped by the external urethral sphincter, a thin muscular layer that allows voluntary control of urine flow (Figure 1) [8]. The rectourethralis muscle plays a vital role in the positioning of the rectum in relation to the membranous urethra. This structure serves as a connection between the rectum and the urethra in males. It is an extension of the longitudinal muscle layer of the rectum and is located above the perineal body, where the cavernous nerve and venous plexus are situated [9].

However, there is a dearth of specific data about pelvic anatomy, such as the distances between the rectourethral space and neighboring structures, which are essential to train surgeons on different approaches (abdominal, transanal, perineal) to surgery for rectal cancer and avoid urethral injuries in these complex procedures. Thus, we aimed to provide a comprehensive understanding of the anatomical region with the highest risk of urethral injury during rectal cancer surgery to improve surgical planning.

To this end, we undertook a three-phase study consisting of cadaveric dissection, retrospective analysis of imaging studies, and retrospective observational analysis of surgical cases.

## 2. Materials and Methods

### 2.1. Study Design and Ethical Considerations

This study was carried out in collaboration between the departments of colorectal surgery and diagnostic imaging at our hospital and the department of anatomy at the medical school with which it is affiliated.

The participating institutions’ ethics committee approved the study protocol (No. 0485762), confirming that all aspects of the study conform to the principles of the Helsinki agreement. In the first phase of the study, we followed all applicable local and international ethical guidelines and laws related to the use of human cadaveric donors in anatomical research [10].

In the second and third phases of the study, the ethics committee waived the need for informed consent due to the retrospective nature of the study and null impact on the care that the patients received.

### 2.2. Anatomical Critical Point Description

Figure 1 illustrates the anatomy of the male pelvis in sagittal (Figure 1A) and axial sections (Figure 1B), showing the relations between the urethra, prostate, rectourethral muscle, sacrum, and walls of the rectum.

### 2.3. Cadaveric Dissection

We obtained 9 adult male cadaveric pelvises free of proctological disease or surgical damage to the rectum from the university’s body donation program. We processed 1 pelvis to obtain semi-thin transversal slices (1500 µm), which were included using the Biodur^®^P40 (BiodurTM, Heidelberg, Germany) plastination technique [11]. We sectioned the remaining 8 pelvises (3 formalin-fixed and 5 cryopreserved) in a mid-sagittal plane, inserting a transanal device (GelPOINT ^®^ Path Transanal Access Platform, Applied Medical Resources Corp., Rancho Santa Margarita, CA, USA) and a Foley catheter into 3 of the cryopreserved pelvises before sectioning to determine their effects on the anatomical relationships among the structures.

To describe the critical point of urethral injury, we took the following measures (Figure 2):-The shortest distance between the anterior surface of the rectum and the membranous urethra (R-Mu distance);-The distance between the point on the anterior wall of the rectum closest to the membranous urethra and the anal margin (R-Am distance);-The angle formed by the intersection of the axis of the anal canal and the axis of the rectum (anorectal angle).

After taking the measurements, we dissected the pelvises meticulously, noting the morphological characteristics of the tissue components of the supralevator space and paying special attention to the rectoprostatic septum and the relations in the area between the anterior face of the rectum and the membranous urethra.

### 2.4. Analysis of Imaging Studies and Clinical Cases

We retrospectively analyzed consecutive cases from the prospectively maintained database in our tertiary care department of male patients with tumors located <10 cm from the anal verge who underwent curative elective rectal cancer surgery other than transanal minimally invasive surgery or local resection between January 2016 and January 2023 (Figure 3).

We analyzed the same measures described previously in pelvic magnetic resonance imaging (MRI) (Philips Healthcare, Andover, MA, USA) (Figure 4). A radiologist blinded to patients’ clinical information used dedicated software (Starviewer 13.3; UdG, Girona, Spain) to measure the R-Mu distance, the R-AM distance, and the anorectal angle, as defined above, on T2-weighted and diffusion-weighted pelvic MRI studies.

To characterize the patients, we recorded their age, body mass index, tumor height and tumor stage assessed through MRI, surgical approach, incidence of urethral injury, and postoperative morbidity, classifying complications as minor (Clavien–Dindo grade I–II) or major (Clavien–Dindo grade III–IV) [12].

### 2.5. Statistical Analysis

We used descriptive statistics to summarize the patients’ demographic, clinical, and anthropometric characteristics. We expressed categorical variables as absolute and relative frequencies and continuous variables as means and standard deviations or medians and ranges, as appropriate. We used statistical package SPSS v. 20.0 (SPSS Inc., Chicago, IL, USA) for all analyses.

## 3. Results

### 3.1. Cadaveric Dissection

Due to the challenges encountered in achieving a precise midsagittal cut that would section the urethra and rectum at their exact midpoint, only four cases (three cryopreserved pelvises and one formalin-fixed pelvis) were considered valid for distance measurements.

The mean R-Mu distance was 1.18 cm (range, 0.8–2), the mean R-Am distance was 2.64 cm (range, 2.1–3), and the mean anorectal angle was 126.4° (range, 116–150) (Table 1).

In all the pelvises, dissection of the supralevator space identified celluloadipose tissue in the space between the anterior wall of the rectum and the posterior face of the lower urinary bladder and prostate. In this space, we were able to isolate the rectoprostatic fascia (Denonvilliers’ fascia), a membranous extension attached distally to the posterior surface of the prostate.

In the distal area, at the point where the rectum and membranous urethra are closest to one another, we were able to observe and to isolate a fibromuscular tissue with transversal bundles, coinciding with the description of the perineal body and rectourethral muscle (Figure 5). The observation of this area in the semi-thin transversal section showed a fibrous tissue at the midline corresponding to the perineal body. Bilaterally to it, lower-density areas that include muscular fibers could be identified as representation of rectourethral muscle. At this level, muscular fibers from the longitudinal layer of the rectum wall and from the puborectalis muscle joined the rectourethral muscle, thus contributing to the formation of it (Figure 6).

In the three specimens in which the transanal device and Foley catheter had been inserted, the correlation of the anatomy encountered during transanal surgery was better. The images show the complexity of placing the transanal device in the cadaveric pelvis, and how the orientation of the device relative to the anorectal angle varied with the position of the device within the anal canal, the length of the anal canal, and the anchoring of the device in relation to the levator muscles (Figure 7).

### 3.2. Analysis of Imaging Studies

The flowchart in Figure 3 outlines the selection of patients for this phase of the study. Between January 2016 and January 2023, a total of 549 patients underwent radical rectal cancer surgery; urethral injury occurred in 4 (0.73%). We analyzed the data from 244 consecutive men with tumors located <10 cm from the anal verge (median age, 67 years (IQR, 59–76); 54 (22.1%) obese; 216 (88.5%) ASA III or IV; 193 (82.1%) stage T3–T4 and 195 (82.9%) stage N1 or N2; 34 (13.9%) with synchronous metastases). The tumor height from the anal verge was 5 cm to 10 cm in 131 (53.7%) and ≤5 cm in 113 (46.3%); neoadjuvant therapy was administered in 211 (86.5%) (Table 2).

On preoperative MRI, a total of 229 images (93.9%) were analyzed. The median R-Mu distance was 1 cm (IQR, 0.7–1.2; range, 0.2–2.3), the median R-Am distance was 4.3 cm (IQR, 3.8–5; range, 2–7.3), and the median anorectal angle was 128° (IQR, 121–137; range, 87–160) (Table 1).

Of 244 patients, laparoscopy was performed in 99 (40.6%) patients, robot-assisted surgery in 73 (29.9%), TaTME in 63 (25.8%), and open surgery in 9 (3.7%). In 17 (7.2%) patients, the procedure was converted into open surgery. The median operative time was 248 min (IQR, 200–300). Major complications (≥IIIb Clavien–Dindo) were observed in 21 (8.6%) patients. Urethral injury occurred in 4 (1.64%) patients: in 2 (4%) of the 50 patients who underwent abdominoperineal resection and in 2 (3.2%) of the 63 patients who underwent TaTME. The median hospital stay was 7 days (IQR, 5–10.8) (Table 3).

Due to the small sample size (*n* = 4) and heterogeneity of the cases, we cannot draw conclusions (Table 4).

### 3.3. Analysis of Clinical Cases

In the four cases of urethral injury, the diagnosis was established intraoperatively. The injury was repaired with a 3–0 or 4–0 braided suture alongside a urinary catheter, and the outcome was assessed by voiding cystourethrography (VCUG) for 4 weeks after the procedure. In two cases, repair was successful; the catheter was removed and no further complications were observed. In one case, VCUG revealed a leak in the membranous urethra, and the catheter was repositioned under cystoscopy guidance; 4 weeks later, VCUG showed no leaks from the membranous urethra, and the catheter was removed. In the remaining case, when the patient sought immediate medical attention for urinary catheter-related discomfort, the catheter was replaced without imaging or cystoscopic guidance, leading to a significant complication that necessitated suprapubic catheterization. Additionally, in this patient with a history of radiation therapy and a grade 3 fistula, a perineal approach was utilized to perform gracilis muscle interposition along with an oral mucosa graft.

## 4. Discussion

The current report synthesizes the findings from an anatomical study in cadavers, an analysis of imaging cases, and an analysis of clinical cases to investigate factors contributing to the risk of urethral injury in rectal cancer surgery in men and to determine the critical distances where this complication can occur. Clinicians need to be aware of the potential for urethral injury, a rare complication of surgery for rectal cancer [3] that is usually due to traumatic urethral catheterization [13].

In the overall scope of our series, the incidence rate of urethral injury was 0.73%. However, in high-risk patients (men with middle and distal rectal lesions), the incidence was 1.64%. When analyzing by techniques, the incidence of urethral injury was 3.2% in TaTME and 4% in abdominoperineal resection. These rates are higher than those reported in other studies, where the reported incidence was calculated including all patients, regardless of risk level. The median R-Mu and R-Am distances measured in the cadaveric pelvises differed greatly from those measured in MRI. However, the small size and heterogeneity of the sample in the cadaveric study preclude generalizable conclusions.

By focusing on specific techniques in high-risk patients, our study reveals that the risk of urethral injury is higher than what might be surmised from a general survey of the relevant literature. Our approach targets the group of patients most vulnerable to this complication, thus enabling us to establish appropriate safety measures and explore alternative, less risky options [14].

The risk of injuring other urological structures during rectal cancer surgery is much lower; the incidence of bladder injury and ureteral injury are 0.15 and 0.06 per cent, respectively [4].

The learning curves for surgical techniques are a topic of ongoing debate and vary widely among publications [15,16,17]. Given the increased incidence of urethral injury during the period when surgeons are learning the TaTME technique, various safety measures have been recommended, including using urethral catheters with built-in light and indocyanine green, and structured learning programs with regular supervision through proctorship programs [18,19].

The success of surgical correction of rectourethral fistulas depends heavily on the stage of the fistula and the appropriate selection of the initial surgical treatment. The International Collaborative Study on urethral injury with the TaTME approach found that urethral repair failed, necessitating permanent urinary diversion in 9% of patients; moreover, even in cases of successful repair, 18% of patients had permanent urinary dysfunction [2].

Urethral injuries are usually diagnosed intraoperatively when the urinary catheter is exposed or in the immediate postoperative period when urine leakage is observed. If the edges of the lesion are viable and can be closed without tension, the best option is primary repair with a 16–18 FG silicone catheter. Repair should be verified by serial voiding cystourethrography 4 to 6 weeks after the procedure. Factors associated with the failure of primary repair include radiation therapy, previous pelvic surgeries, and infections. If primary closure is impossible or leakage persists beyond 8 weeks, repair options include using healthy tissue (e.g., the interposition of a gracilis muscle flap with an oral mucosa graft [20], radical prostatectomy, or cystoprostatectomy, tailoring management to patients’ medical history and expectations, as well as other pertinent factors [6,21].

Through the exchange of insights and concerns among surgeons, radiologists, and human anatomists, this multidisciplinary research has provided anatomic references for surgery that can have a profound impact on surgical strategies and help reduce the risk of complications [22].

Due to the difficulties in acquiring specimens, the sample in the cadaveric phase of our study was limited to nine cases; a larger sample might have ensured more reliable information. The analyses of MRI and surgical cases were retrospective and conducted at a single center; prospective multicenter studies would likely have yielded more generalizable results. Nevertheless, combining the information from the three phases of the study has enabled us to provide valuable data that can help reduce the risk of urethral injury during rectal cancer surgery.

## 5. Conclusions

Urethral injuries in men undergoing abdominoperineal resection or TaTME for rectal cancer are uncommon but devastating. The critical point for membranous urethral injury is situated at a depth of 2 cm to 7.3 cm from the anal margin, where the distance between the rectum and the membranous urethra measures 0.2 cm to 2.3 cm.

Improving surgeons’ knowledge of anatomy through collaboration with anatomists and radiologists fosters better multidisciplinary teamwork, facilitates training opportunities outside the live patient environment, improves the interpretation of MRI within complex anatomical spaces, and increases safety during surgical procedures.

## Figures and Tables

**Figure 1 cancers-15-04955-f001:**
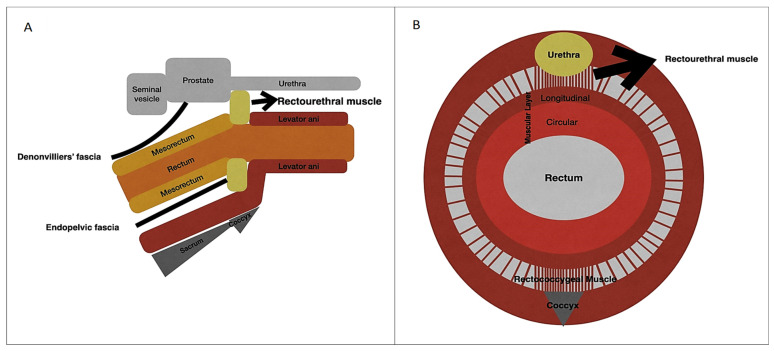
Diagram illustrating the anatomy of the male pelvis in sagittal (**A**) and axial sections (**B**).

**Figure 2 cancers-15-04955-f002:**
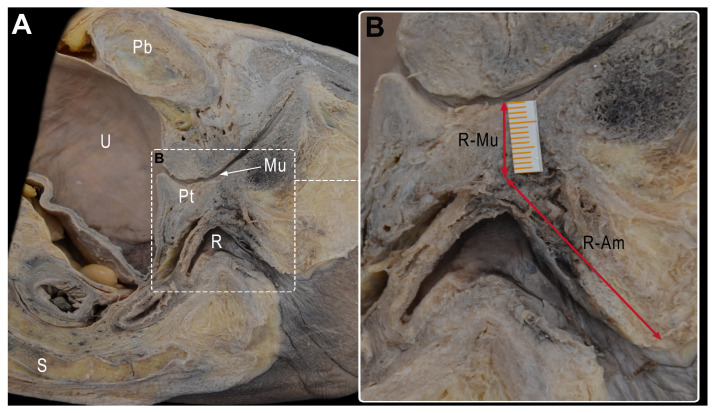
(**A**) Medial view of the left side of a sagittal sectioned pelvis. (**B**) Magnification of rectourethral area with measurements of R-Mu distance between the anterior surface of the rectum and the membranous urethra and R-Am distance between the point on the anterior wall of the rectum closest to membranous urethra and the anal margin. Pb: pubis; U: urinary bladder; Pt: prostate; Mu: membranous urethra; R: rectum; S: sacrum.

**Figure 3 cancers-15-04955-f003:**
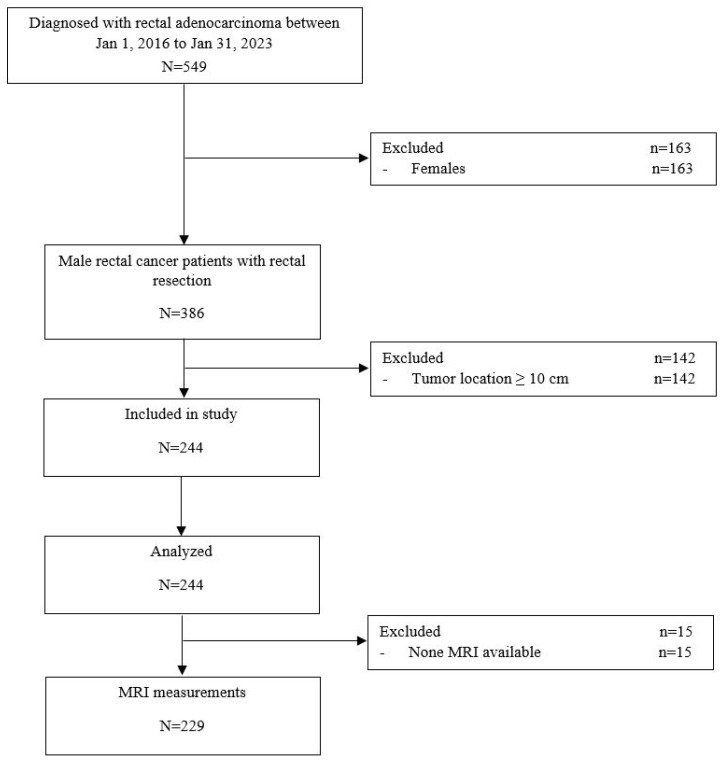
Study flowchart illustrating the process of patient selection for the study.

**Figure 4 cancers-15-04955-f004:**
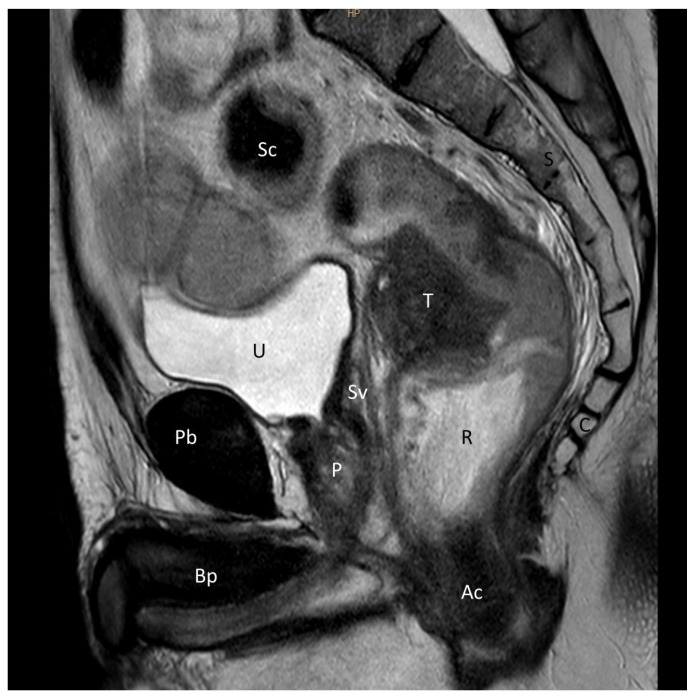
Sagittal T2-weighted MRI view of a male pelvis showing rectal cancer located 6 cm anteriorly to the left. The cancer was staged as T3N2aM0. Pb: pubis; U: urinary bladder; Bp: Body penis; P: Prostate; Sv: seminal vesicle; R: rectum; S: sacrum; Sc: sigmoid colon; T: tumor; C: coccyx; Ac: anal canal.

**Figure 5 cancers-15-04955-f005:**
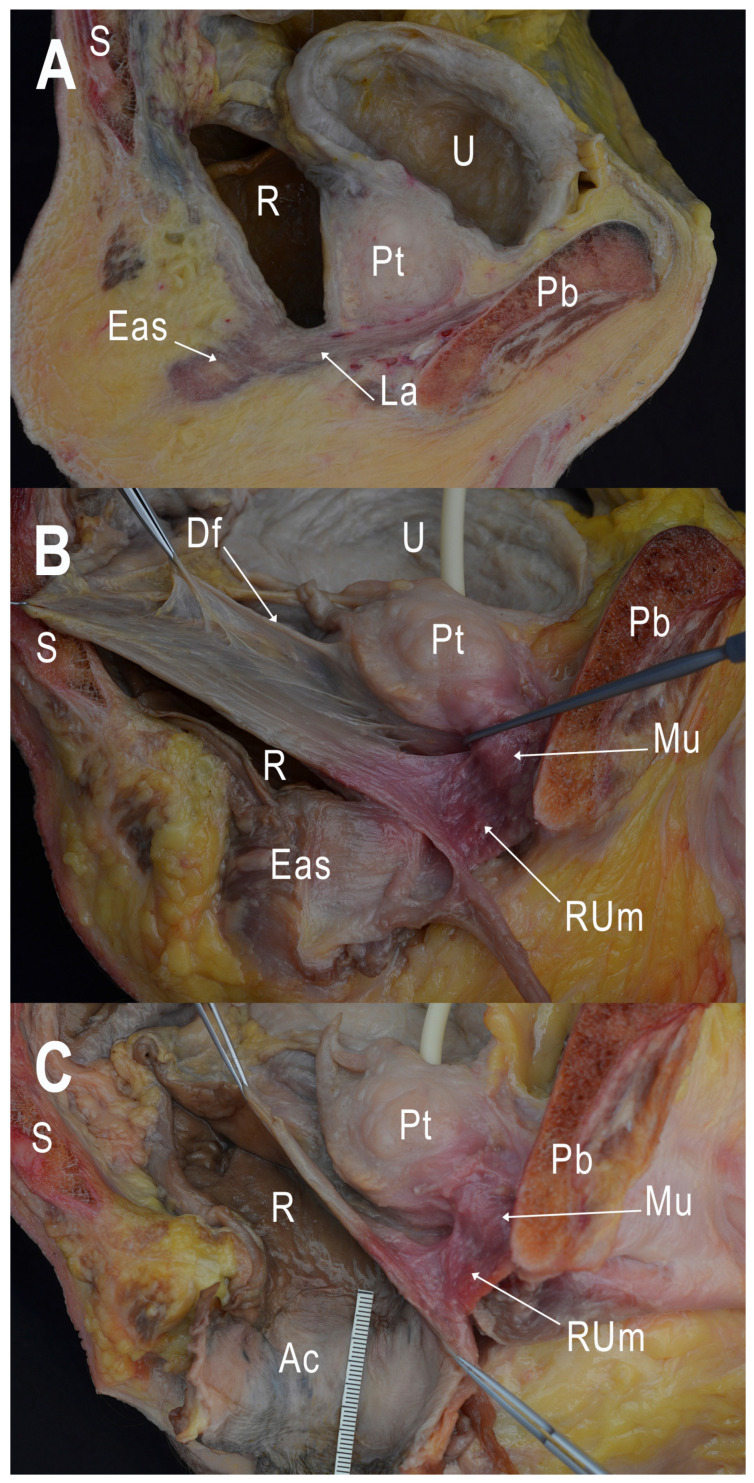
Dissection of the medial view of the left side of a sagittal sectioned pelvis. (**A**) Sagittal section before dissection. (**B**) Dissection of the supralevator space. Identification of Denonvilliers’ rectovesical septum and fibromuscular transversal tissue bundles of the rectourethral muscle. The external anal sphincter around the anal canal is maintained. (**C**) Exposition of the anal canal after removing of the external anal sphincter. Pb: pubis; S: sacrum; U: urinary bladder; Pt: prostate; R: rectum; La: levator ani muscle; Eas: external anal sphincter; Df: Denonvilliers’ fascia; Mu: membranous urethra; Rum: rectourethral muscle; Ac: anal canal.

**Figure 6 cancers-15-04955-f006:**
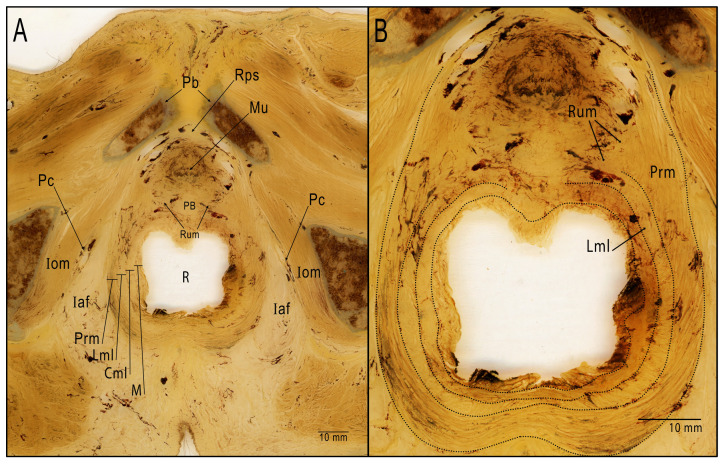
Semi-thin transversal section (1500 µm) of a male pelvis showing the relationship between the anterior wall of the rectum and the membranous urethra. (**A**) We can observe the perineal body and the rectourethral muscle beside it. (**B**) Magnification of the rectourethral area. Identification of the rectourethral muscle and its relationship with the layers of the rectum and puborectalis muscle. Pb: pubis; Rps: retropubic space; Mu: membranous urethra; PB: perineal body; Rum: rectourethral muscle; R: rectum; M: mucosa of rectum; Cml: circular muscular layer; Lml: longitudinal muscular layer of rectum; Prm: puborectalis muscle; Iaf: ischioanal fossa; Iom: internal obturator muscle; Pc: pudendal canal.

**Figure 7 cancers-15-04955-f007:**
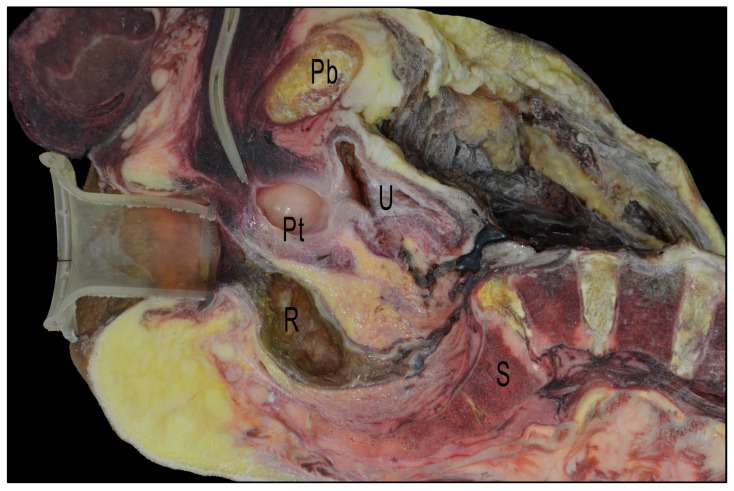
Sagittal section of a cryopreserved cadaveric pelvis with GelPOINT^®^ Path Transanal Access Platform and a urinary catheter inserted. Pb: pubis; U: urinary bladder; Pt: prostate; R: rectum; S: sacrum.

**Table 1 cancers-15-04955-t001:** Measurements in cadaveric pelvises and magnetic resonance images (MRIs).

	Anatomic Pelvic Measurements					
Cadaveric Pelvises (N = 8)	R-Mu Distance (cm)		R-Am Distance (cm)		Anorectal Angle (°)	
Formalin-fixed (N = 3)						
Case 1	2		3		116	
Case 2	NA		NA		NA	
Case 3	NA		NA		NA	
Cryopreserved (N = 5)						
Case 4	0.8		3		116	
Case 5	NA		NA		NA	
Case 6 (Foley catheter + GP)	0.9		2.7		120	
Case 7 (Foley catheter + GP)	1.2		2.1		130	
Case 8 (Foley catheter + GP)	1		2.4		150	
NA; not available, GP: GelPOINT ^®^ Path Transanal Access Platform						
MRI pelvic measurements						
	R-Mu distance (cm)		R-Am distance (cm)		Anorectal angle (°)	
	Median (IQR)	Mean ± SD	Median (IQR)	Mean ± SD	Median (IQR)	Mean ± SD
MRI (N = 229)	1 (0.7–1.2)	1 ± 0.44	4.3 (3.8–5)	4.39 ± 0.96	128 (121–137)	128 ± 12.3

**Table 2 cancers-15-04955-t002:** Description of demographic and clinical characteristics of the patient samples in the observational study.

	Observational Study
Variables	N = 244
BMI, (kg/m^2^)	
<30	190 (77.9%)
≥30	54 (22.1%)
Age (years)	
<70	141 (57.8%)
≥70	103 (42.2%)
ASA	
I or II	28 (11.5%)
III or IV	216 (88.5%)
Tumor height, (cm)	
>10–15	-
≥5–10	131 (53.7%)
0–5	113 (46.3%)
cT- MRI	
cT1	3 (1.3%)
cT2	36 (15.3%)
cT3	163 (69.4%)
cT4	33 (14.0%)
cN- MRI	
cN0	40 (17.0%)
cN1	83 (35.3%)
cN2	112 (47.7%)
Synchronous metastases	
No	210 (86.1%)
Yes	34 (13.9%)
Neoadjuvant treatment	
No	33 (13.5%)
Yes	211 (86.5%)

BMI; Body mass index, ASA; American Society of Anesthesiologists physical status classification, MRI; magnetic resonance imaging.

**Table 3 cancers-15-04955-t003:** Surgical procedures and outcomes in the observational study.

	Observational Study
Variables	N = 244
Surgical approach	
Laparoscopy	99 (40.6%)
Robot-assisted	9 (3.7%)
TaTME	63 (25.8%)
Open	73 (29.9%)
Conversion to open surgery	
No	218 (92.8%)
Yes	17 (7.2%)
Ureteral injury	
No	240 (98.4%)
Yes	4 (1.6%)
Duration of surgery, in min, median (IQR)	248 (200–300)
Hospitalization, in days, median (IQR)	7 (5–10.8)
Morbidity (Clavien–Dindo)	
None	156 (63.9%)
<IIIb	67 (27.5%)
≥IIIb	21 (8.6%)

**Table 4 cancers-15-04955-t004:** Characteristics of patients with urethral injuries.

	Age	Tumor Location (cm)	BMI (kg/m^2^)	Tumor Stage	Neoadjuvant Therapy	Time Interval (Weeks)	Intervention
Patient 1	53	9.3	18.42	T3N1M0	Yes	6.1	APR
Patient 2	57	5	31.43	T3N2M0	Yes	5.7	APR
Patient 3	57	5.5	22.04	T3N1M0	Yes	7.1	TaTME
Patient 4	63	3.1	26.84	T3N1M0	Yes	5.4	TaTME

APR, abdominoperineal resection.

## Data Availability

The data presented in this study are available upon request from the corresponding author. The data are not publicly available due to ethical regulations.

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
