# Peer review of "Urethral Injury in Rectal Cancer Surgery: A Comprehensive Study Using Cadaveric Dissection, Imaging Analyses, and Clinical Series"

_cancers, 2023, doi:10.3390/cancers15204955_

Round 1

Reviewer 1 Report

Urethral injury is a significant risk in the transanal approach, and I think the discussion in this paper is interesting and clinically meaningful. On the other hand, I have the impression that the data presented are too anatomically biased to be published in this journal. I recommend that this paper be submitted to an anatomical journal.

Was the APR done under direct vision?

What are the characteristics of the four cases with urethral injuries?

Author Response

Urethral injury is a significant risk in the transanal approach, and I think the discussion in this paper is interesting and clinically meaningful. On the other hand, I have the impression that the data presented are too anatomically biased to be published in this journal. I recommend that this paper be submitted to an anatomical journal.

Thank you for your interesting comments; they have helped us improve our manuscript. All changes are highlighted in yellow in the revised manuscript.

Was the APR done under direct vision?

Thank you for your comment. The APR is performed in two stages, an abdominal stage and a perineal stage. The abdominal stage is performed using minimally invasive surgery, while the perineal stage is performed under direct vision.

What are the characteristics of the four cases with urethral injuries?

Thank you for pointing this out. We have included Table 4 with patient characteristics in the results section in the revised manuscript.

Reviewer 2 Report

It is difficult to understand the usefulness of a cadaveric study when the AA. comment on that  the small size and heterogeneity of the sample  preclude generalizable conclusions. The same consideration   are valid when the number of urethral lesions is fortunately low : probably the real risk factor appears to be pre-operative radiotherapy: or this reason it might be interesting to take into consideration the length of the interval between RT and surgery

Author Response

It is difficult to understand the usefulness of a cadaveric study when the AA. comment on that  the small size and heterogeneity of the sample  preclude generalizable conclusions. The same consideration   are valid when the number of urethral lesions is fortunately low : probably the real risk factor appears to be pre-operative radiotherapy: or this reason it might be interesting to take into consideration the length of the interval between RT and surgery

Thank you for your comments; it has greatly contributed to the enhancement of our manuscript.

All modifications have been highlighted in yellow in the revised manuscript.

As you pointed out, in order to study the critical urethral injury zone, it is essential to conduct an anatomical study. This allows us to assess the distances between different structures. A more thorough understanding of anatomy is associated with safer surgery and a reduced incidence of intra- or postoperative complications.

As you mentioned, the rate of urethral injury is low, but it is higher than that described in the literature, which is why we decided to undertake this study. Although it is an infrequent complication, it is exceedingly serious when it does occur.

In Table 4, we have included the time interval between radiotherapy and surgery.

Thank you once again for your valuable input, and we appreciate your careful review of our manuscript.

Reviewer 3 Report

Regarding the strong points, this research focuses on a rare topic in the anatomical structure of the male urethra and its relation to improving surgical planning. The overview of the article is well-organized and easily understandable. The historical background and rationale are clear and reasonable. The authors highlight the importance of the risk of urethral damage during rectal cancer surgery, emphasizing the potential for significant short-term complications and long-term effects and the research questions are clear. The anatomical descriptions of the membranous urethra, external urethral sphincter, and rectourethralis muscle provide readers with a clear understanding of the relevant anatomy. In methods, the detailed procedures are well mentioned. Illustrating the anatomical measurement is clear and understandable with various figures indicating the possibility of reproducibility of the method. The results provide a detailed description of three-phase study including cadaveric dissection, analysis of imaging studies, and retrospective analysis of surgical cases. This approach allows for a comprehensive examination of the topic and is consistent with the objectives of the study. The authors provide well conclusion of the main finding, the usefulness of the study, and the limitations. The comprehensive results are completely discussed including incidence data, safety measures, and treatment options for urethral injuries. For the weak points, some parts of the article show different measurement units as millimeters and centimeters. Using the same measurement unit makes the reader better understand and not be confused. The cadaveric dissection results are limited by the small sample size due to challenges in achieving a precise midsagittal cut. This limitation restricts the findings. The authors briefly mention factors associated with the failure of primary repair such as radiation therapy, previous pelvic surgeries, and infections, but do not additionally describe how these factors influence outcomes or provide guidance on risk assessment.

Author Response

Regarding the strong points, this research focuses on a rare topic in the anatomical structure of the male urethra and its relation to improving surgical planning. The overview of the article is well-organized and easily understandable. The historical background and rationale are clear and reasonable. The authors highlight the importance of the risk of urethral damage during rectal cancer surgery, emphasizing the potential for significant short-term complications and long-term effects and the research questions are clear. The anatomical descriptions of the membranous urethra, external urethral sphincter, and rectourethralis muscle provide readers with a clear understanding of the relevant anatomy. 

In methods, the detailed procedures are well mentioned. Illustrating the anatomical measurement is clear and understandable with various figures indicating the possibility of reproducibility of the method.

The results provide a detailed description of three-phase study including cadaveric dissection, analysis of imaging studies, and retrospective analysis of surgical cases. This approach allows for a comprehensive examination of the topic and is consistent with the objectives of the study. The authors provide well conclusion of the main finding, the usefulness of the study, and the limitations.

The comprehensive results are completely discussed including incidence data, safety measures, and treatment options for urethral injuries.

Thank you for your comments and for evaluating the article as well organized and easily understandable. All changes are highlighted in yellow in the revised manuscript.

For the weak points, some parts of the article show different measurement units as millimeters and centimeters. Using the same measurement unit makes the reader better understand and not be confused. 

Thank you for the suggestion; we have standardized all units of measurement to centimeters in the revised manuscript.

The cadaveric dissection results are limited by the small sample size due to challenges in achieving a precise midsagittal cut. This limitation restricts the findings.

Thank you for pointing this out. As you mentioned, the results regarding the cadavers are limited, unfortunately due to the high cost of cadavers, we were only able to study 9 pelvises.

The authors briefly mention factors associated with the failure of primary repair such as radiation therapy, previous pelvic surgeries, and infections, but do not additionally describe how these factors influence outcomes or provide guidance on risk assessment.

Thank you for your interesting comment, three out of the four patients had erectile dysfunction during the follow-up; however, since we did not assess the erectile function of patients without urethral injury, we cannot attribute this dysfunction to the urethral injury.

Round 2

Reviewer 1 Report

I've reviewed the revised manuscript and am satisfied with the corrections made. Thank you for addressing the previous comments.

Reviewer 2 Report

 I thought for a long time before commenting on this. I think that at this point the work can be accepted for publication. even if doubts remain such as the role of the surgical  team, as independent variable'